# Wnt Pathway-Targeted Therapy in Gastrointestinal Cancers: Integrating Benchside Insights with Bedside Applications

**DOI:** 10.3390/cells14030178

**Published:** 2025-01-24

**Authors:** Anirudh Nayak, Hannah Streiff, Ivan Gonzalez, Oluwabomi Oluwatomi Adekoya, Itzcoatl Silva, Anitha Kota Shenoy

**Affiliations:** Master of Science in Biomedical Sciences Program, California Health Sciences University, Clovis, CA 93612, USA; nayak2756@chsu.edu (A.N.); streiff2694@chsu.edu (H.S.); gonzalez2754@chsu.edu (I.G.); adekoya2696@chsu.edu (O.O.A.); silva2752@chsu.edu (I.S.)

**Keywords:** gastrointestinal cancers, Wnt signaling pathway, β-catenin, clinical trials, colorectal cancer, pancreatic cancer, gastric cancer, hepatocellular carcinoma

## Abstract

The Wnt signaling pathway is critical in the onset and progression of gastrointestinal (GI) cancers. Anomalies in this pathway, often stemming from mutations in critical components such as adenomatous polyposis coli (APC) or β-catenin, lead to uncontrolled cell proliferation and survival. In the case of colorectal cancer, dysregulation of the Wnt pathway drives tumor initiation and growth. Similarly, aberrant Wnt signaling contributes to tumor development, metastasis, and resistance to therapy in other GI cancers, such as gastric, pancreatic, and hepatocellular carcinomas. Targeting the Wnt pathway or its downstream effectors has emerged as a promising therapeutic strategy for combating these highly aggressive GI malignancies. Here, we review the dysregulation of the Wnt signaling pathway in the pathogenesis of GI cancers and further explore the therapeutic potential of targeting the various components of the Wnt pathway. Furthermore, we summarize and integrate the preclinical evidence supporting the therapeutic efficacy of potent Wnt pathway inhibitors with completed and ongoing clinical trials in GI cancers. Additionally, we discuss the challenges of Wnt pathway-targeted therapies in GI cancers to overcome these concerns for effective clinical translation.

## 1. Introduction

Cancer is a complex and multifaceted disease characterized by the uncontrolled proliferation of transformed cells, which possess the capacity to invade surrounding tissues, metastasize to distant organs, and adapt through natural selection [1,2]. Cancer is the second leading cause of death in the US and among the three leading causes of death worldwide [3,4]. According to “Cancer Statistics”, in 2024, it is projected that approximately 2,001,140 new cancer cases will be diagnosed in the US. Gastrointestinal (GI) cancers, which include esophageal, stomach, liver, pancreatic, and colorectal cancers (CRCs), collectively are expected to be among the leading causes of cancer-related deaths [5,6]. CRC is the most commonly diagnosed and deadliest form of GI cancer, followed by pancreatic cancer, which has a dismal five-year survival rate of only 13% in the US. Other significant contributors include liver, stomach, and esophageal cancers [5].

GI cancers, including colorectal, gastric, and pancreatic cancers, are notorious for their high rates of resistance to conventional therapies and frequent relapse following treatment [7,8,9,10,11]. The conventional therapy for GI cancers includes surgery, chemotherapy, radiation, and targeted therapy directed towards EGFR, HER2, and VEGF [12,13]. Immunotherapy targeting immune checkpoint inhibitors, such as PD-1, PDL-1, and CTLA-4, have recently gained traction in treating GI cancers including pancreatic, gastric, esophageal, biliary tract, hepatocellular, and colorectal cancers [14,15,16,17]. Despite the recent advances in the treatment of GI cancers, the development of drug resistance, often driven by the activation of alternative signaling pathways, mutations in key oncogenes, and the presence of cancer stem cells, limits the long-term efficacy of these current therapies [7,8,9,10,11,13,18,19,20]. These challenges urge the need for novel therapeutic strategies that can overcome resistance mechanisms and prevent disease recurrence.

The cancer development involves the dysregulation of several key cellular processes, including proliferation, differentiation, DNA repair, and apoptosis [21]. At the molecular level, these processes are controlled by various signaling networks, many of which become abnormally activated or suppressed in cancer [22]. Among these signaling pathways, the Wnt signaling pathway has gathered significant attention due to its vital role in various physiological aspects of cellular function and its frequent dysregulation in numerous cancers, including GI cancer [23,24,25,26,27,28,29,30,31,32,33,34]. The Wnt signaling pathway is well known to be involved in maintaining the normal function of the intestinal epithelium, and genetic alterations in key components APC and β-catenin have been linked to the development of CRC [31,32,35,36], gastric cancer [27,37], and other malignancies within the GI tract [29,38]. Furthermore, the Wnt signaling pathway contributes to therapeutic resistance in various GI cancers, including gastric, liver, and colorectal cancers [31,39,40,41,42,43,44]. Thus, targeting this pathway, which is crucial in GI cancer pathogenesis and resistance, is the logical approach to treating these cancers. Developing combination therapies that inhibit Wnt signaling, alongside other targeted approaches, holds promise for improving patient outcomes and overcoming resistance in GI cancer treatments.

The Wnt signaling pathway has been extensively researched and explored in the context of gastrointestinal (GI) cancer treatment. However, no Wnt pathway-targeted therapies have been successfully implemented in clinical practice thus far. This review aims to provide an overview of the Wnt signaling pathway, its critical role in various GI cancers, and its potential to be targeted in GI cancer treatments. Furthermore, we summarize and discuss completed and ongoing clinical trials to date to accentuate both the challenges and opportunities associated with integrating Wnt-targeted therapies in combination with other treatment approaches for GI cancer. Unlike many reviews that focus exclusively on either preclinical mechanisms or clinical outcomes, this work bridges the gap by combining the findings from both domains. By addressing preclinical data and the current state of ongoing and completed clinical trials, this review highlights the clinical potential of targeting Wnt pathway components and thoroughly integrates benchside molecular insights into bedside therapeutic applications regarding gastrointestinal cancers. Each clinical trial is meticulously analyzed, detailing both outcomes and the underlying reasons for success or failure. Through this thorough integration of findings, critical information gaps are filled, and targeted Wnt therapies are positioned as one of the most promising approaches to potentially address poor patient outcomes in aggressive gastrointestinal cancers.

## 2. Wnt Signaling Pathway

The Wnt signaling pathway, which is highly conserved across species, is crucial in embryogenesis, cell differentiation and development, tissue repair, and many other processes [45,46,47,48,49]. The intracellular signaling cascade of the Wnt pathway is initiated by secreted lipid-modified proteins from the Wnt family, comprising nineteen proteins [50,51]. Wnt ligands bind to Wnt receptors, such as frizzled (FZD), receptor tyrosine kinase-like orphan receptor (ROR), receptor-like tyrosine kinase (Ryk), protein tyrosine kinase 7 (PTK7), muscle-specific kinase (MuSK), and co-receptor LDL-receptor-related proteins (LRPs) on the surface of target cells [52,53,54]. This interaction triggers intracellular signaling events that ultimately lead to cellular responses, which are critical in regulating key biological processes. Wnt signaling is broadly classified into two main branches: the canonical (β-catenin-dependent) and non-canonical (β-catenin-independent) pathways, each characterized by distinct mechanisms and functions involving different Wnt ligands and specific receptor families [55]. To fully understand the scope and complexities of the Wnt signaling pathway, it is crucial to explore the unique mechanisms and functions of these two pathways.

### 2.1. Canonical Wnt Signaling Pathway

The canonical Wnt signaling pathway, also called the Wnt/β-catenin signaling pathway, is critical in cell differentiation, proliferation, maturation, and generating the correct body axis specifications [56,57]. The Wnt1 class ligands that include Wnt2, Wnt3, Wnt3a, and Wnt8 are involved in the canonical Wnt signaling pathway that binds to specific receptors known as the frizzled (FZD) receptors and the coreceptors LDL-receptor-related proteins 5 and 6 (LRP5 and LRP6), leading to activation of the pathway [58,59,60].

In the absence of Wnt ligands, β-catenin is continuously degraded by a cytoplasmic protein complex known as the destruction complex. This complex is composed of adenomatous polyposis coli protein (APC), AXIN, casein kinase 1 (CK1), and glycogen synthase kinase-3 (GSK-3), along with E3-polyubiquitin ligase β-TrCP and protein phosphatase 2A (PP2A) [61]. CK1 first phosphorylates β-catenin at the Ser45 residue, initiating a “priming” process that enables GSK-3 to phosphorylate the Ser33, Ser37, and Thr41 residues. This phosphorylation creates a binding site for β-TrCP, which acts as an adaptor protein, facilitating the polyubiquitination and subsequent proteasomal degradation of β-catenin [60,62].

In the presence of the Wnt ligands, their binding to FZD receptors initiates the phosphorylation of LRP5/6, promoting the interaction between both receptors and activating Disheveled (DVL). Activated DVL inhibits the destruction complex, resulting in the accumulation of β-catenin in the cytoplasm. β-catenin is then translocated into the nucleus, where it binds to T cell factor/lymphoid enhancer factor (TCF/LEF) transcription factors, driving the expression of target genes such as c-Myc, cyclin D, and AXIN2 [62] (Figure 1). These Wnt-regulated genes play crucial roles in cellular differentiation and mitotic progression [63].

Several endogenous activators and inhibitors regulate the Wnt/β-catenin signaling pathway. Secreted proteins R-spondin (RSPO) and Norrin activate the canonical Wnt signaling pathway, each utilizing distinct mechanisms. RSPO, a family of secreted proteins, potentiates Wnt signaling by binding to leucine-rich repeat-containing G-protein-coupled receptors (LGRs), specifically LGR4, LGR5, and LGR6, which then recruit and stabilize the Wnt co-receptors FZD and LRP5/6 on the cell surface. This stabilization occurs through the inhibition of the E3 ubiquitin ligases RNF43 and ZNRF3, which normally downregulate Wnt receptors, thereby enhancing pathway activity [64,65]. On the other hand, Norrin activates canonical Wnt signaling uniquely by binding directly to FZD-4, bypassing the need for Wnt ligands and interacting with the LRP5/6 co-receptor. This interaction promotes β-catenin stabilization, and its subsequent translocation to the nucleus initiates the transcription of Wnt target genes. While RSPO is essential for maintaining intestinal homeostasis, Norrin plays a significant role in retinal development and vascular stability [66,67,68,69,70].

Secreted proteins, such as secreted frizzled-related proteins (SFRPs), Wnt inhibitory factor (WIF), and Dickkopf-related protein 1 (DKK1), play crucial roles in modulating Wnt activity by inhibiting pathway signaling [71]. SFRPs, including SFRP1, 2, 4, and 5, act by directly binding to Wnt ligands, preventing their interaction with frizzled receptors and thus blocking the initiation of Wnt signaling. WIF also functions similarly by binding to Wnt proteins, effectively sequestering them and inhibiting their ability to activate the pathway [72]. DKK1, on the other hand, primarily interferes with the Wnt/β-catenin (canonical) pathway by binding to LRP5/6 co-receptors, thereby preventing their interaction with the frizzled receptors [73]. Moreover, some sFRPs and Dkks do not inhibit Wnt function, making their role in Wnt signaling a subject of debate [71]. These antagonists play a key regulatory role in maintaining balance within the Wnt signaling pathway. They are critical in preventing aberrant signaling, which is often linked to cancer development and progression.

### 2.2. Non-Canonical Wnt Signaling Pathway

The non-canonical Wnt pathways are characterized by their β-catenin independence, which does not involve β-catenin stabilization or nuclear translocation. Instead, these pathways utilize diverse mechanisms, including the Wnt/Ca^2+^ pathway and the Wnt/PCP (Planar Cell Polarity) pathway, with distinct signaling routes to achieve functions such as cellular polarization and migration (Figure 1) [74]. Additionally, the non-canonical pathways utilize different ligands from the Wnt5a family (Wnt 4, Wnt5a, Wnt5b, Wnt6, Wnt7a, and Wnt11) that bind to different receptors [58].

In the Wnt/Ca^2+^ pathway, Wnt binds to FZD receptors, activating phospholipase C (PLC) and thus increasing intracellular calcium levels [60]. This increase in calcium leads to the activation of calcium-dependent proteins such as calcineurin, calmodulin-dependent kinase II (CAMKII), and cell–division cycle 42 (CDC42) [74]. The activation of these effector proteins initiates gene transcription via the nuclear factor of activated T-cells (NFAT) and induces cytoskeletal rearrangements [54,74].

Unlike the Ca^2+^ pathway, the PCP pathway utilizes co-receptors in addition to FZD, such as protein tyrosine kinase 7 (PTK7), muscle-specific kinase (MuSK), receptor tyrosine kinase-like orphan receptor (ROR1/ROR2), and receptor-like tyrosine kinase (Ryk) [54]. After binding of the ligand, the phosphorylation of DVL leads to the activation of several effector proteins such as Dvl-associated activator of morphogenesis (DAAM), Ras homolog gene-family member A (RHOA), RHO-associated coiled-coil-containing protein kinase (ROCK), RAC1 protein, and c-Jun-N-terminal kinase (JNK). These proteins, via an intracellular cascade, all support actin polymerization, thus contributing to cell polarity and migration [54,60,74].

## 3. GI Cancers and Wnt Signaling as Therapeutic Targets

Building on the understanding of GI cancers, exploring the role of the Wnt signaling pathway in each type of GI malignancy is essential. Anomalous Wnt signaling plays a significant role in tumor initiation, progression, and therapy resistance in colorectal, gastric, pancreatic, and esophageal cancers. In the following sections, we will review the involvement of the Wnt pathway in the pathogenesis of these cancers and discuss potential drugs targeting Wnt signaling based on the individual location or organ within the body (Figure 2). Furthermore, we will summarize the advancements and challenges in the completed and ongoing clinical trials that utilize the Wnt signaling pathway or its components as a therapeutic target for GI cancers.

### 3.1. Esophageal and Gastric Cancers

Despite the decline in new diagnoses of esophageal and stomach cancers over recent years, the estimated number of new cases of these cancers ranks third highest among all GI cancers [5]. This signifies the burden these cancers pose and, thus, the need for continued research and improved therapeutic strategies.

In esophageal squamous cell carcinoma (ESCC) and esophageal adenocarcinoma (EAC), abnormal activation of the Wnt pathway has been linked to tumorigenesis and poor prognosis. For instance, studies have shown that WNT2, secreted by tumor-associated fibroblasts, drives tumor cell proliferation and invasion, particularly in WNT2-positive ESCC cases, which are associated with aggressive disease and lymph node metastasis [75]. Elevated β-catenin expression, common in esophageal carcinoma, is a potential biomarker for disease progression [76]. Activation of this pathway also contributes to epithelial–mesenchymal transition (EMT), enhancing cell invasion, while its inhibition has been shown to sensitize ESCC cells to chemoradiotherapy [28,77]. NEK2 has also been identified as a driver of ESCC cell proliferation via Wnt/β-catenin signaling, correlating with poor survival, highlighting its role as a potential target [78]. The complex interaction between Wnt and other pathways like PI3K/AKT further potentiates the need for targeted therapies to disrupt these signaling networks.

Constitutive activation of the Wnt/β-catenin signaling pathway accounts for 30–50% of gastric adenocarcinomas [27,79]. This activation often results from mutations in key components such as APC and β-catenin, leading to unchecked cell growth and tumorigenesis. A study examining 311 gastric cancer samples found that 29% exhibited nuclear β-catenin localization, with mutations in exon 3 of the β-catenin gene detected in about 26% of those tumors, highlighting a significant link between β-catenin mutations and gastric cancer progression [80]. One of the studies showed that 46 out of 52 samples of stomach adenocarcinoma, a common type of gastric cancer, displayed nonsynonymous mutations in APC genes [81]. RUNX3 acts as a tumor suppressor by forming a complex with β-catenin and TCF4, inhibiting their transcriptional activity [27]. RUNX3 has been demonstrated to suppress and activate the Wnt signaling pathway by interacting with the TCF4/β-catenin complex in gastric cancer cells [82]. Moreover, mutations in RNF43, an E3 ubiquitin ligase that regulates Fzd receptor levels on the cell surface [83], are found in gastric tumors [84]. These mutations enhance Wnt signaling by stabilizing Fzd receptors, thereby promoting tumor progression [84].

In addition to aberrations in canonical Wnt signaling, non-canonical Wnt signaling has also been shown to play a role in the pathogenesis of gastric cancer. Wnt5a, a ligand associated with the non-canonical pathway, has been implicated in promoting the invasion and migration of gastric cancer cells. It modulates integrin adhesion turnover, facilitating cell movement through the extracellular matrix. While Wnt5a can act as a tumor suppressor in some contexts, its overexpression is correlated with aggressive tumor behavior in gastric cancer [85]. DVL, a scaffolding protein essential to canonical Wnt signaling, is also a significant effector in the non-canonical Wnt signaling pathway [86,87]. DVL mediates the adhesion-related effects of Wnt5a in gastric cancer cells to regulate cytoskeleton dynamics, further enhancing cell motility and tumor progression [85]. Epigenetic regulation further complicates the role of the Wnt pathway in gastric cancer. The loss of inhibitors like sFRP (secreted frizzled-related protein) and DKK1 (Dickkopf-related protein 1) due to epigenetic modifications can lead to aberrant Wnt signaling, often accompanied by hypermethylation of gene promoters that contribute to gastric carcinogenesis [88,89,90].

Research has also identified significant roles for porcupine (PORCN) and DKK1 in GI cancers, particularly esophageal and gastric cancers, emphasizing their potential as therapeutic targets. Porcupine is an enzyme responsible for the palmitoylation of Wnt proteins, a process crucial for their secretion and activation of the Wnt signaling pathway [91]. Preclinical studies have shown that inhibiting porcupine can effectively disrupt Wnt signaling, thereby reducing tumor growth in GI cancers [92]. For instance, studies have indicated that pharmacological inhibition of Wnt/β-catenin signaling can sensitize esophageal cancer cells to chemoradiotherapy [77], suggesting a synergistic effect when combined with porcupine inhibitors. Furthermore, inhibition of porcupine with IWP-2 in gastric cancer cell lines has been shown to decrease cell proliferation and increase apoptosis, suggesting that targeting this enzyme could be an effective strategy for treating gastric tumors reliant on Wnt signaling [93].

These encouraging preclinical results have led to the initiation of phase 1 clinical trial testing of two porcupine inhibitors, LGK974 (WNT974) and CGX1321, in advanced GI cancers, including esophageal and gastric cancers (Table 1, Figure 2). LGK974 (WNT974) was evaluated as a monotherapy in Wnt-dependent advanced solid tumors, including GI cancers. Results from the study showed that LGK974 (WNT974) as monotherapy was generally well tolerated. Biomarker analyses from the trial indicated that LGK974 (WNT974) may also influence immune cell recruitment to tumors and potentially enhance the efficacy of checkpoint inhibitors [94]. Consequently, a second arm of the study was designed to investigate LGK974 (WNT974) in combination with the anti-PD-1 monoclonal antibody, spartalizumab (PDR001), along with further immune signature analyses (NCT01351103). However, results from this combination study are yet to be released.

The phase 1 clinical trial involving CGX1321 in advanced GI cancers had an estimated completion date of 2021 with 39 estimated enrollments. However, results have not been posted thus far, and the trial status remains unknown (NCT03507998). In another phase 1 clinical trial, CGX1321 was also evaluated in combination with agents encorafenib, cetuximab, and immune modulator pembrolizumab for various GI cancers (NCT02675946). This multicenter, open-label study was conducted in two phases. Phase 1 included a CGX1321 single-agent dose escalation phase in advanced GI tumors, a CGX1321 single-agent dose dpansion phase for advanced GI tumors, and a roll-over cohort combining CGX1321 with pembrolizumab for patients who progressed on CGX1321 alone. Phase 1b investigated CGX1321 with pembrolizumab in advanced CRC and with encorafenib and cetuximab for BRAFV600E-mutated CRC. Both phases aimed to assess safety, pharmacokinetics, and clinical activity. Although intended to conclude in 2023, the last update was in 2022, with 72 estimated enrollments and no results posted. Of the twenty-five US sites and one in Taiwan, only two had completed enrollment, with most US locations still recruiting or yet to start.

While porcupine activates the Wnt signaling pathway, DKK1 is a Wnt pathway antagonist. DKK1 was initially characterized as a tumor suppressor but has also been implicated in promoting tumor growth and metastasis. Elevated expression of DKK1 correlates with a poor prognosis in a range of cancers, including patients with esophageal and gastric cancer. A study found that high serum levels of DKK1 correlated with unfavorable five-year survival rates and could serve as a prognostic marker for risk stratification in clinical settings [95]. Similarly, another study in serum from patients with gastric cancer had high levels of DKK1 being associated with poorer overall survival [96]. Furthermore, a chimeric adenovirus-mediated DKK1 overexpression strategy was employed to inhibit aberrantly activated Wnt/β-catenin signaling in gastric cancer stem cells, which resulted in reduced cell viability, anchorage-independent colony formation, and invasion, indicating its potential tumor-suppressive role [97]. This suggests that targeting DKK1 or modulating its activity could improve treatment outcomes.

Two clinical trials evaluating the DKK1 inhibitor DKN-01 in advanced solid tumors that include GI cancer (NCT04681248) and gastric or gastroesophageal cancers (NCT04363801) are currently available and active, respectively (Table 1, Figure 2). The trial on advanced solid tumors is an intermediate-size expanded access protocol (EAP) (NCT04681248) (Table 1, Figure 2). Patients who were receiving DKN-01 in a parent study (NCT05480306: Phase 2 clinical trial with a combination of DKN-01 and standard chemotherapy for CRC) will be allowed to continue in this EAP along with the combination chemotherapy, and patients who are DKN-01 naïve with advanced solid tumors approved by the treating oncologist will receive DKN-01 as monotherapy. In two other clinical trials: a phase 2 clinical trial (NCT04363801), where DKN-01 is used in combination with the anti-PD-1 monoclonal antibody, tislelizumab ± chemotherapy and a phase 1/2 clinical trial (NCT04166721), where DKN-01 is used in combination with the anti-PD-L1 monoclonal antibody, atezolizumab trial in patients with gastric or gastroesophageal cancer. The estimated completion of the trial is 2025 (Table 1, Figure 2).

Given its crucial role as the final mediator of Wnt pathway activation, directly targeting β-catenin offers a rational and potentially more effective approach for combating subsets of gastric cancers driven by aberrant Wnt signaling. By inhibiting β-catenin, the core of the signaling cascade can be disrupted, addressing the oncogenic effects seen in these cancers. Phase 1/2 clinical trial with β-catenin inhibitor, FOG-001, is recruiting patients with metastatic or locally advanced solid tumor, including those with GI cancers (NCT05919264) (Table 1, Figure 2). The estimated completion of the trial is 2027.

### 3.2. Liver Cancer

The regenerative properties of hepatocytes, the cell type in the liver, make the liver an organ with one of the highest recovery rates in the case of acute injury [98]. The Wnt signaling pathway contributes to liver regeneration [99]; however, errors in this pathway can lead to several adverse outcomes, including cancer [52,100]. Current research suggests that hepatocellular carcinoma (HCC), the most common type of primary liver cancer;cholangiocarcinoma (a type of bile duct cancer); hepatoblastoma (a rare liver cancer in children); and several other rare tumors that present in the liver may arise due to the irregular activation of the Wnt/β-catenin signaling pathway [101,102,103]. Atypical activation of Wnt/β-catenin signaling has been observed in precancerous liver lesions and cancerous growths, highlighting the importance of understanding the nuances within this pathway for treating liver cancer [104].

Viral infections, including hepatitis B (HBV) and hepatitis C (HCV), are significant contributors to liver cirrhosis and HCC [105,106,107]. The oncogenic potential of HBV is primarily linked to the integration of viral DNA into the host genome, resulting in genomic instability and the subsequent accumulation of precancerous lesions. HBV-induced carcinogenesis is further driven by mutations in viral oncogenes that modulate the Wnt signaling pathway [108,109,110,111,112]. Specifically, HBV infection can lead to the upregulation of Wnt ligands, such as WNT1 and WNT3, frizzled receptors like FZD2 and FZD7, and the suppression of Wnt antagonists, including secreted frizzled-related proteins SFRP1 and SFRP5, in chronic HBV cases [108,109,110,112,113,114]. On the other hand, chronic HCV infection also predisposes to HCC, but its mechanism is mediated through the hyperactivation of the viral core protein and nonstructural proteins NS3 and NS5A [115,116]. This dysregulation enhances the expression of Wnt ligands (WNT1, WNT3A), frizzled receptors, and LRP5/6 while simultaneously leading to the downregulation of Wnt antagonists like SFRP2 and Dickkopf 1 (DKK1) via hypermethylation of their promoter regions [117,118,119]. Furthermore, activation of FZD10 in liver cancer stem cells (CSCs) through METTL3-dependent N6-methyladenosine methylation enhances self-renewal, tumorigenicity, and metastasis via the FZD10-β-catenin/YAP1 axis, which is linked to drug resistance, particularly to lenvatinib, and suggests that targeting this pathway could improve treatment outcomes in HCC [120]. In a preclinical study, a drug named PRI-724 (active drug: C-82) that inhibits the interaction between β-catenin and its coactivator CREC-binding protein (CBP) was shown to inhibit proliferation and increase apoptosis in β-catenin-activated HCC [121]. Tegavivint is a novel inhibitor that interferes with β-catenin and transducin β-like protein 1 (TBL1) interaction, preventing the cell’s ability to promote cellular proliferation, and has shown promising results as an anti-tumor agent in desmoid tumors and osteosarcomas [122]. Following a phase 1 trial to determine the maximum tolerated dose (MTD) and toxicity, a phase 2 clinical trial to define the anti-tumor activity of Tegavivint (BC2059) is planned in pediatric patients with varied tumors with Wnt pathway aberration, including HCC (NCT04851119). Another clinical phase 1/2 clinical trial (NCT05797805) is ongoing in advanced HCC, where initially, the maximum tolerated dose (MTD) will be established using Tegavivint as a single agent. After the dose escalation phase is completed, two selected dose levels will be expanded for further evaluation to determine the optimal dose before finalizing the recommended phase 2 dose (RP2D) for monotherapy dose expansion. In the event of sufficient clinical benefit, the study’s second phase will explore the combination of Tegavivint with pembrolizumab in patients with mutations in the CTNNB1 or AXIN1 genes who have previously been treated with a PD-1/PD-L1 inhibitor [123] (Table 2, Figure 3).

Cholangiocarcinoma, or bile duct cancer, is often diagnosed at an advanced stage, leading to a poor prognosis for many patients. Despite a rising incidence of this malignancy, progress in treatment options has been limited [126]. This cancer affects the epithelial biliary lining of extra- and/or intrahepatic bile ducts. Several clinical and preclinical studies have alluded to atypical activation of Wnt/β-catenin signaling as a key driver in its initiation and progression [127,128,129,130]. Notably, several frizzled ligands and WNT7B and WNT10A are overexpressed in the advanced stages of cholangiocarcinoma [126]. In addition, the Wnt ligand secretion protein porcupine was upregulated, as confirmed by PCR arrays [126]. Furthermore, disruptions in this pathway contribute to multidrug resistance by inducing the expression of P-glycoprotein efflux pumps on cancer cells [131]. While surgical resection remains the primary treatment option for early-stage cholangiocarcinoma, its effectiveness is often hindered by frequent recurrences and late-stage diagnosis [132]. Based on the role of the Wnt signaling pathway in pathogenesis and therapy resistance, the pathway components serve as attractive targets in the treatment of cholangiocarcinoma. Combining a porcupine inhibitor with a PI3K inhibitor yielded promising results in the EGI-1 cholangiocarcinoma cell line [133]. Furthermore, RXC004, an oral porcupine inhibitor, was shown to block tumor growth and reverse immune evasion in Wnt ligand-dependent cancer models [134], indicating that targeting the porcupine to inhibit the Wnt signaling pathway could be a potential therapeutic approach for this cancer.

Building on these findings, RXC004, an oral porcupine inhibitor, is currently undergoing phase 1 clinical trials (NCT03447470) for the treatment of biliary tract cancers and other solid tumors, both as a monotherapy and in combination with an anti-PD-1 monoclonal antibody, Nivolumab (Table 2, Figure 3). RXC004 has shown a favorable pharmacokinetic profile, effectively inhibiting Wnt ligand palmitoylation, secretion, and subsequent pathway activation [134]. Moreover, it has exhibited antiproliferative effects in colorectal and pancreatic cell lines dependent on Wnt ligands, especially those with RNF43 mutations or RSPO3 fusions, further supporting the potential of Wnt pathway inhibition as a therapeutic strategy for cholangiocarcinoma and other Wnt-driven malignancies [134].

### 3.3. Pancreatic Cancer

Pancreatic cancer is a highly volatile cancer as it presents with a lot of complications, including a five-year survival rate of about 13% [5] and a relapse in tumor development in 80% of individuals who took treatment-resistance measures [5,135]. It is crucial to note how efforts in proactivity and the stage at which this cancer is detected heavily play a role in a patient’s outcome. Pancreatic cancer is one of the most lethal cancers, as it is detected extremely late in its progression and, therefore, reduces the patient’s opportunity for surgery [136,137]. Additionally, pancreatic cancer is known to have unsuccessful responses to certain chemotherapies [136,137]. Although significant therapeutic breakthroughs are awaited in pancreatic cancer treatment, there has been a significant advancement in understanding the Wnt signaling pathway, which plays a critical role in the development and progression of the disease. Wnt signaling plays a crucial role in pancreatic cancer progression through multiple mechanisms, mainly via the canonical β-catenin-dependent pathway [135,138,139]. Experimental studies have demonstrated elevated Wnt ligands and receptor levels in pancreatic ductal adenocarcinoma (PDAC) cells, with aberrant activation promoting tumor cell proliferation, invasion, and stem cell maintenance [140,141,142]. A genome-wide CRISPR screening approach identified a druggable Wnt-FZD5 signaling circuit as a critical vulnerability specific to RNF43-mutant pancreatic tumors, revealing a potential therapeutic strategy for this genetically defined subset of pancreatic cancers [143]. The inhibition of Wnt signaling using specific antagonists has been shown to reduce pancreatic cancer cell growth and metastatic potential in both in vitro and in vivo models [139,144,145].

Numerous studies in cancer models have shown that certain GSK3β inhibitors can increase tumor sensitivity to chemotherapeutic agents [146,147,148]. In both in vitro and in vivo studies, pharmacologic inhibition of GSK3β with 9-ING-41, also known as Elraglusib, has been shown to disrupt gemcitabine-induced DNA repair processes, promoting apoptosis in chemoresistant pancreatic ductal adenocarcinoma (PDAC) cells [149]. Currently, a multi-institutional, open-label, four-arm phase 2 study (NCT05077800) is examining the efficacy of combining FOLFIRINOX (Folinic Acid, 5-Fluorouracil, Irinotecan, and Oxaliplatin) with 9-ING-41, a GSK-3β inhibitor, and Losartan, a TGF-β blocker, in treating adults with previously untreated metastatic pancreatic adenocarcinoma (PAC) (Table 2, Figure 4). This trial aims to determine if this approach can restore or enhance the effectiveness of standard chemotherapy in patients for whom it has been ineffective or only marginally effective. This study is still recruiting and is expected to conclude in 2025. A separate phase 2 clinical trial (NCT03678883) is currently underway to assess the safety and efficacy of 9-ING-41, both as a standalone treatment and in combination with cytotoxic agents, in patients with previously untreated advanced PDAC (Table 2, Figure 4). Early findings indicate that 9-ING-41, combined with gemcitabine and nab-paclitaxel (GnP), shows promising clinical activity against advanced PDAC while maintaining an acceptable safety profile [124]. A randomized phase 2 study has also been initiated to compare the efficacy of 9-ING-41, administered either once or twice weekly with GnP, versus GnP alone in patients with previously untreated metastatic or locally advanced pancreatic cancer.

Another GSK3β inhibitor, LY2090314, has been demonstrated to regulate the intrinsic PDAC cell resistance with key chemotherapeutic agents, such as gemcitabine, oxaliplatin, nab-paclitaxel, and Irinotecan [150]. Preclinical studies have also shown that drug combinations with LY2090314 decrease PDAC cell viability, while mouse models treated with LY2090314 and nab-paclitaxel exhibited improved overall survival and reduced cytotoxicity in non-malignant pancreatic cells [151]. LY2090314 was also investigated in a phase 1 dose-escalation study alongside pemetrexed and carboplatin in patients with advanced or metastatic solid tumors. Although intended to be included as per the study plan, no patients with pancreatic cancer were enrolled (NCT01287520) (Table 2, Figure 4). The initial safety profile of LY2090314 was established in this trial; however, its efficacy in combination with pemetrexed and carboplatin requires confirmation in larger, randomized trials [125].

Lastly, a phase 1 clinical trial (NCT01302405) investigating PRI-724, a β-catenin/CREC-binding protein (CBP) inhibitor, was conducted to establish the maximum tolerated dose (MTD) in solid tumors, including PAC. Once identified, the plan was to escalate doses of PRI-724 in patients with CRC in combination with a modified FOLFOX6 regimen to determine the MTD of the combined treatment. Unfortunately, despite no safety issues, the study was terminated due to low enrollment. However, another phase 1 clinical trial (NCT01764477) was conducted following this clinical trial, where PRI-724 was used in combination with gemcitabine in subjects with advanced or metastatic PAC eligible for second-line therapy after failing first-line therapy with FOLFIRINOX (or FOLFOX) (Table 2, Figure 4). Results are unavailable despite the successful completion of this trial.

### 3.4. Small Intestinal Cancer

Wnt signaling plays an essential role in small intestinal development, particularly in regulating the proliferation, differentiation, and maintenance of intestinal stem cells (ISCs), which are crucial for the continuous renewal of the intestinal epithelium [26]. Activation of Wnt signaling upregulates ISC markers such as Lgr5, supporting ISC proliferation within the crypts of Lieberkühn, which in turn sustains the epithelial layer [152]. Disruption of Wnt signaling highlights its critical function: studies show that mice deficient in the Wnt effector gene Tcf4/Tcf7l2 lack crypts and display only differentiated villus cells, accentuating the importance of Tcf4 for ISC function and crypt maintenance [153].

Conversely, hyperactivation of the Wnt signaling pathway, often through mutations in the APC gene or the constitutive activation of β-catenin, can lead to abnormal epithelial cell proliferation and polyp formation, implicating Wnt dysregulation in the early stages of small intestinal cancer [154]. The APC gene is a tumor suppressor gene that regulates Wnt signaling by acting as a scaffold for the destruction complex that targets β-catenin for proteasomal degradation [33]. This deregulation is a common mechanism driving tumorigenesis, as continuous Wnt activation promotes adenoma formation and is frequently seen in small intestinal malignancies [155]. Further supporting this, studies involving the introduction of the Wnt inhibitor DKK1 lead to the loss of crypts and significant changes in cell differentiation patterns [156,157]. Furthermore, DKK1 has been shown to repress β-catenin/TCF target gene expression and CD44 expression, specifically in the ileum, emphasizing its role in modulating the Wnt pathway in distinct intestinal regions [157].

Given the role of the Wnt signaling pathway in both ISC maintenance and tumorigenesis, therapeutic targeting of Wnt pathway components in preclinical small intestinal cancer models has shown promise. For example, β-catenin inhibitors have effectively reduced tumorigenicity and cell proliferation in small intestinal adenocarcinoma models [158]. Tankyrase inhibitors such as XAV939, which stabilize AXIN (a negative Wnt regulator), have also effectively reduced Wnt-driven tumor growth in preclinical settings, suggesting potential therapeutic avenues [158]. Furthermore, targeting the RSPO-LGR5 interaction disrupts ISC renewal, providing another promising target for intervention in small intestinal cancers [64]. Moreover, deletion of the Wnt target gene Myc is critical for crypt formation after birth and to regulate crypt size and proliferation in adult mice, emphasizing the influence of the Wnt pathway on ISC maintenance and epithelial renewal [159]. Additionally, the deregulation of Myc has been shown to promote tumor progression, and it could be a potential target in intestinal cancer treatment [160].

Supported by preclinical evidence of Wnt signaling pathway targeting in small intestinal cancer and leveraging broader findings in GI cancers, a phase 1/2 clinical trial with the β-catenin inhibitor FOG-001 is now recruiting patients with locally advanced or metastatic solid tumors, including small intestinal cancers (NCT05919264) (Table 1, Figure 2). Exisulind, an antineoplastic agent and inhibitor of phosphodiesterase (PDE) isozymes PDE5 and PDE4, has been shown to attenuate β-catenin and induce apoptosis [161]. Exisulind was intended for evaluation in a phase 2/3 clinical trial to prevent the development and growth of duodenal polyps in patients with FAP (NCT00026468), a condition that can eventually lead to duodenal cancer. However, based on the available information, the clinical trial may have been withdrawn for administrative reasons unrelated to clinical outcomes or the trial’s conduct.

### 3.5. Colorectal Cancer

Wnt signaling abnormalities are the primary cause of cancer development in CRC. Mutations in Wnt pathway components result in unchecked cell division and the development of tumors. Gaining insight into the complexities of Wnt signaling in CRC will be necessary for creating tailored treatments and enhancing patient outcomes. The most common factor in CRC is truncating mutations in the adenomatous polyposis coli protein (APC) gene, found in 70–80% of all sporadic colorectal tumors [33,162]. The inactivation of APC leads to the activation of the canonical pathway and key regulators of mitotic progression and cellular differentiation, such as cyclin D and c-Myc [32,162,163]. Aberrations in the APC gene that lead to nonsense and missense mutations are considered the initial event in nearly all CRCs, suggesting that Wnt hyperactivity is the critical driver of most CRC incidences [31,162]. Familial adenomatous polyposis (FAP) is a hereditary condition caused by germline mutations in the APC gene, resulting in the development of hundreds to thousands of adenomatous polyps in the colon and rectum, often during adolescence or early adulthood [164]. If left untreated, these polyps have an almost 100% likelihood of progressing to CRC due to the unchecked activation of Wnt signaling and subsequent cellular dysregulation [165]. Besides APC, AXIN2 is found to be mutated in about 5% of CRC cases. AXIN2 mutations, particularly a recurring frameshift mutation, result in a truncated AXIN2 protein that stabilizes β-catenin and activates Wnt signaling, as it serves as a scaffolding protein in the destruction complex similar to APC [163]. Although not as frequently found in CRCs, mutations in the CTNNB1 gene, which encodes β-catenin, can also lead to hyperactivation of the Wnt signaling pathway via β-catenin accumulation [166]. Other than the activation of Wnt signaling, mutations in CTNNB1 have also been shown to affect E-cadherin-mediated cell–cell adhesion, which could impact the metastatic spread of cancerous cells [167]. RSPOs can also contribute to CRC, accounting for 4–10% of cases [168]. RSPO enhances the Wnt/β-catenin signaling pathway by auto-ubiquitinating and degrading ZNRF3/RNF43 [169]. Mutations in ZNRF3/RNF43 or overexpression/translocation of RSPO2 can lead to enhanced Wnt signaling and, thus, uncontrolled cell growth and tumor progression [163,170].

In CRC, the Wnt signaling pathway is a strategic target as it plays a consequential function in cell survival, proliferation, and differentiation [171]. The most popular and effective treatment methods among the many available options include focusing on porcupine, β-catenin, and the Wnt pathway itself [31,172]. Additionally, combination therapies work synergistically to enhance anti-tumor immune responses while inhibiting the Wnt pathway [172]. Repurposed drugs are also emerging as a possible treatment, with reduced risks and distinct advantages in the ability to understand mechanisms and side effects succinctly [171]. These approaches to targeting the Wnt/β-catenin pathway draw attention to the strategic points of intervention, offering a meticulous therapeutic strategy for combating CRC.

LGK974, a first-in-class porcupine inhibitor, has been shown to downregulate the downstream Wnt-activated signaling, thus abolishing Wnt-driven cancer tumors [173]. LGK974, or WNT974, is currently in an active phase 1 clinical trial specifically targeted for advanced solid tumors, including CRC (NCT01351103) (Table 1, Figure 5). As discussed earlier under the esophageal and gastric cancer section, a recent proof of concept study showed encouraging results, with the drug being generally well tolerated and analyses showing inhibition of Wnt signaling [94]. The anti-cancer activity of LGK974 against BRAFV600-mutant KRAS wild-type metastatic CRC harboring RNF43 mutations or RSPO was evaluated in a phase 1/2 trial (NCT02278133) (Table 3, Figure 5). The trial enrolled 20 subjects and tested for and measured the incidence of dose-limiting toxicities of the drug in phase 1 and the overall response rate in phase 2. Though the trial is completed, the results still need to be made available. Another drug targeting porcupine, RXC004, in combination with the anti-PD-1 antibody Nivolumab, has recently completed phase 2 trials in iRNF43- or RSPO-mutated, metastatic, microsatellite-stable CRC who have progressed following the therapy with the current standard of care (NCT04907539) (Table 3, Figure 3). The results are yet to be released. CGX132, another porcupine inhibitor, is in phase 1 clinical trials for various GI cancers, including CRC (NCT02675946, NCT03507998). The details of the trials are discussed under the esophageal and gastric cancer section (Table 1, Figure 5). A phase 2 clinical trial (NCT03678883) is currently underway to assess the safety and efficacy of 9-ING-41, both as a standalone treatment and in combination with cytotoxic agents, in patients with advanced cancers, including colon cancer (Table 2, Figure 5).

**Table 3 cells-14-00178-t003:** Clinical trials of drugs targeting Wnt signaling pathway in FAP and colorectal cancer.

Indication	Drug	Target	Regimen	Clinical Status	Phase	Reference
Colorectal cancer	LGK974	Porcupine	Combination with LGX818 (anti-BRAF) and cetuximab (anti-EGFR)	Completed	Phase 1/2	NCT02278133
Colorectal cancer	RXC004	Porcupine	Monotherapy OR/AND Combination with nivolumab (anti-PD-1)	Completed	Phase 2	NCT04907539
Advanced or metastatic solid tumors, including colorectal cancer	ST316	BCL-9, a coactivator of β-catenin	Monotherapy AND Combination with FOLFIRI and bevacizumab (anti-VEGF) OR Fruquintinib (anti-VEGFR) OR Lonsurf (DNA synthesis inhibitor) and bevacizumab	Recruiting	Phase 1/2	NCT05848739
Colon cancer	Resveratrol	Reduces nuclear levels of β-catenin	Monotherapy	Completed	Phase 1	NCT00256334, [174]
Colorectal cancer	Aspirin	Reduces the expression of Wnt pathway target genes	Monotherapy AND Combination with metformin	Active, not recruiting	Phase 2	NCT03047837
Colorectal cancer	Celecoxib	Reduces the expression of Wnt pathway target genes	Monotherapy	Completed	Phase 2	NCT00582660, [175]
FAP	Niclosamide	Reduces the expression of Wnt pathway target genes	Monotherapy	Completed	Phase 2	NCT04296851
Colorectal cancer	Artesunate	Reduces the expression of Wnt pathway target genes	Monotherapy	Recruiting	Phase 2	NCT02633098
FAP	Azithromycin	APC gene	Monotherapy	Unknown	Phase 4	NCT04454151
FAP	Erythromycin	APC gene	Monotherapy	Unknown	Phase 4	NCT02175914
Colorectal cancer	Lycopene	Reduces the expression of Wnt target genes	Monotherapy in patients treated with panitumumab (anti-EGFR)	Unknown	Phase 2	NCT03167268
Colorectal cancer	Genistein	Wnt signaling pathway	Combination with FOLFOX OR FOLFOX-Avastin	Completed	Phase 1/2	NCT01985763, [176]

Source: *Clinicaltrials.gov*. Data collected on 24 October 2024. Withdrawn and terminated trials are not included.

**Figure 5 cells-14-00178-f005:**
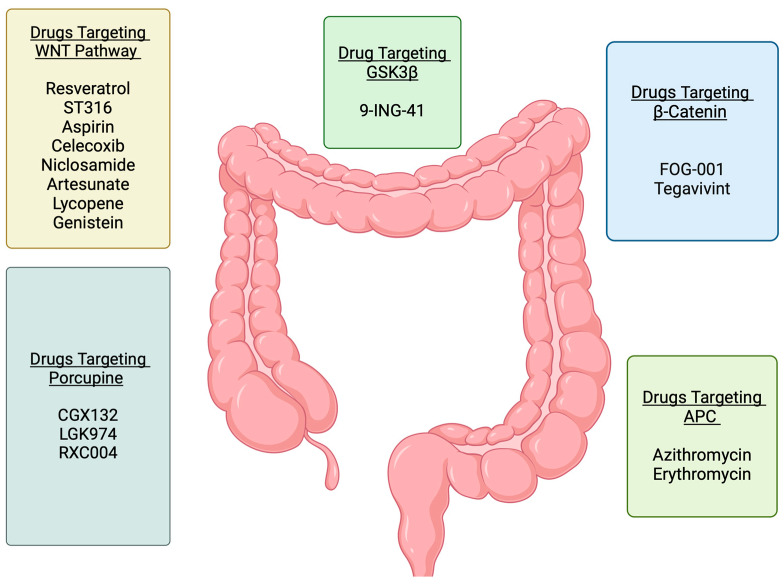
Wnt pathway-targeted drugs in colorectal cancer. The figure illustrates the various Wnt signaling targets and associated drugs that have been or are currently being evaluated in clinical trials for colorectal cancers. Figure created on https://BioRender.com and accessed on 9 December 2024.

Inhibitors targeting β-catenin inhibit its transcriptional activity by disrupting its nuclear accumulation, thus abrogating Wnt signaling and cellular proliferation. Notable examples include PRI-724, Tegavivint, and FOG-001. PRI-724 is a small inhibitory molecule that acts specifically on the interaction between β-catenin and its coactivator, CBP. It inhibits the expression of Wnt target genes that regulate cell division [177]. PRI-724, which was evaluated in a phase 1 clinical trial of pancreatic cancer (NCT01764477) (Table 2, Figure 4), was also evaluated in phase 1 and phase 2 clinical trials for CRC (NCT01302405 and NCT02413853) but were terminated and withdrawn due to low patient enrollment and low drug supply, respectively. Currently, a phase 1/2 clinical trial is underway for Tegavivint, a β-catenin/TBL-1 inhibitor, in patients with recurrent or refractory solid tumors, including CRC and desmoid tumors (NCT04851119) (Table 2, Figure 5). FOG-001, a direct β-catenin inhibitor that blocks its interaction with the TCF transcription factor family, is also recruiting participants in a phase 1/2 trial (NCT05919264) (Table 1, Figure 3).

ST316, currently in phase 2 trials (NCT05848739) (Table 3, Figure 5), is designed to directly inhibit the Wnt signaling pathway by targeting the β-catenin coactivator BCL-9, preventing β-catenin’s translocation to the nucleus. As a first-in-class inhibitor, ST316 disrupts β-catenin signaling, reducing the transcription of Wnt target genes that drive cell proliferation and tumor growth while enhancing apoptosis in CRC cells. By selectively inhibiting oncogenic Wnt signaling, ST316 preserves the physiological functions of β-catenin, making it a promising therapeutic option for Wnt-driven tumors [178]. Additionally, natural compounds like resveratrol, found in grape skin, offer another approach to reducing the Wnt target gene expression; it has completed phase 1 trials (NCT00256334) (Table 3, Figure 5) and demonstrated in vitro efficacy in downregulating Wnt target genes such as cyclinD1 and AXIN2 in normal colonic mucosal cell lines. This finding suggests that dietary supplementation with resveratrol-containing foods can be a potential preventive strategy against colon cancer [174].

Repurposing existing drugs to target the Wnt signaling pathway offers a promising strategy for CRC treatment [179,180]. Leveraging these agents in CRC treatment could accelerate clinical progress, reduce development costs, and provide practical options for targeting this challenging pathway. Non-steroidal anti-inflammatory drugs (NSAIDs), which are COX-2 inhibitors, shown to reduce chronic inflammation, oxidative stress, and Wnt/β-catenin activity, are among the most commonly used repurposed drugs in studies [181]. Aspirin, an NSAID, can inhibit the accumulation of β-catenin in the nucleus, thereby reducing the transcription of Wnt target genes involved in cell proliferation and survival [182]. Aspirin and indomethacin were shown to downregulate the transcriptional activity of β-catenin/TCF-responsive genes, while celecoxib/sulindac reduced β-catenin levels [183,184]. In a recent phase 2 clinical trial (NCT03047837) (Table 3, Figure 3), low-dose aspirin as monotherapy and in combination with metformin was evaluated to prevent CRC after surgery. This trial has been shown to have favorably reduced biomarkers in CRC [175]. Another phase II clinical trial involving patients with CRC received celecoxib, a selective COX-2 inhibitor, for 7 days before undergoing standard resection surgery (NCT00582660) (Table 3, Figure 5).

Other groups of commonly used repurposed drugs are anti-parasitic and antibiotics, which have both been shown to have anti-cancer effects [185]. Niclosamide, an antihelminthic, has been shown to inhibit the Wnt/β-catenin pathway by degrading the membrane receptor LRP6, thus suppressing β-catenin accumulation in the nucleus [186]. The anti-malarial drug artesunate has been shown to suppress the downstream targets of Wnt, thereby promoting cell apoptosis and inhibiting colony formation in uveal melanoma [187]. Artesunate has also been shown to inhibit cell growth in CRC. However, it is by promoting reactive oxygen species-dependent cell senescence and autophagy [188]. Studies using in vivo mouse models demonstrated that oral administration of niclosamide significantly reduced adenoma formation by disrupting AXIN–GSK3 interaction, suggesting that niclosamide could effectively treat patients with FAP [189]. A phase 2 clinical trial using niclosamide against FAP, which is well known to lead to CRC, has been completed, but the trial is under unknown status, and the results are not yet available (NCT04296851) (Table 3, Figure 5). Artesunate is used in a phase 2 trial that is actively recruiting as a pre-operative in stage 2 and 3 CRC (NCT02633098) (Table 3, Figure 3). The aminoglycoside and macrolide antibiotic families have been shown to promote the read-through of nonsense mutations in the APC gene, decreasing oncogenic phenotypes in CRC cells and multiple mouse models [190]. Azithromycin, a macrolide, was examined in ApcMin/+ mice with a truncated APC protein. It was shown to suppress the number and proportions of tumors and downregulation of β-catenin and cyclin D1 [191]. Macrolides, namely azithromycin and erythromycin, were evaluated in phase 4 trials for APC stop codon mutations in FAP, respectively (NCT04454151, NCT02175914) (Table 3, Figure 5). Although the results are not yet available, they could be extremely promising based on the results from the mice study.

The demethylating agent azacitidine has been shown to derepress Wnt target genes in xenograft models, subsequently suppressing tumor growth [192]. In a clinical trial (NCT01882660), decitabine, another demethylating agent, was administered pre-operatively to colon cancer patients to assess Wnt target gene expression. However, no differences were observed in the methylation or expression of Wnt targets [193]. This trial demonstrates that, despite promising preclinical studies, some drugs may fail to translate successfully into clinical practice.

Lycopene, which has been shown to decrease the nuclear levels of β-catenin [194], is assessed in a phase 2 clinical trial (NCT03167268) to mitigate skin toxicity induced by panitumumab in patients undergoing treatment for metastatic CRC (Table 3, Figure 5). Genistein, a soy-derived compound, can inhibit the Wnt signaling pathway [195]. In preclinical studies, genistein has been shown to sensitize the effects of cisplatin on ovarian cancer cells [196] and, when combined with 5-Fluorouracil, synergistically induce apoptosis in chemoresistant cancer cells. In a phase 1/2 trial (NCT01985763), genistein was added to FOLFOX or FOLFOX-Avastin regimens for patients with newly diagnosed stage IV CRC (Table 3, Figure 5). The combination was safe and tolerable, and the efficacy results were notable. This necessitates further investigation into more extensive clinical trials [176].

## 4. Discussion and Conclusions

Targeting the Wnt pathway has emerged as a promising therapeutic strategy across various gastrointestinal (GI) cancers, with numerous clinical trials currently underway to evaluate its efficacy. In this review, we have provided a detailed overview of both benchside research and clinical trials across different GI cancers, highlighting key findings from preclinical studies and ongoing or completed trials. While the Wnt-targeted drugs hold promise in the treatment of various GI cancers, such as gastric, esophageal, liver, and pancreatic cancer, when it comes to small intestinal cancer, the landscape is notably sparse in terms of clinical trials specifically targeting the Wnt pathway. Despite this limitation, benchside research has offered valuable insights, particularly into the therapeutic potential of targeting DKK1, tankyrase, and β-catenin. Preclinical models have demonstrated encouraging results, indicating that inhibitors against these targets could effectively disrupt oncogenic Wnt signaling and potentially improve patient outcomes.

While targeting the Wnt signaling pathway remains a highly promising therapeutic strategy for gastrointestinal (GI) cancers, it is important to acknowledge that, despite numerous clinical trials across different GI malignancies, no Wnt-targeted therapy has yet received FDA approval. One of the reasons for this lack of regulatory success may be attributed to the physiological complexity and ubiquitous role of Wnt signaling, particularly in GI organs, where it plays a critical role in maintaining normal tissue homeostasis, regeneration, and stem cell function. Systemic inhibition of this pathway often leads to significant on-target, off-tumor toxicities, posing a major hurdle in clinical application. Another contributing factor is that, although several promising agents are currently being evaluated in clinical trials, many of these studies are in early phases, with pending results. Also, as discussed earlier, there have also been numerous setbacks in many trials, such as difficulty in recruitment, unpublished results, drug supply, and many others. In this regard, the contribution and quantity of clinical trials are largely unavailable.

To overcome these challenges and to successfully advance through various clinical trial phases, existing strategies must be improved, or new strategies must be designed. One such strategy is to preserve the healthy counterparts and minimize the off-target effects by optimizing the delivery methods and administering the drug locally. The approach of local delivery of Wnt-targeted therapies directly to the tumor site, such as tumor-directed drug delivery systems or localized injections, rather than relying on systemic administration, may allow for higher therapeutic concentrations at the tumor site while minimizing adverse effects on normal tissues.

Consistent with the currently employed strategy, Wnt pathway inhibitors could be more effectively deployed as adjunct therapies alongside standard-of-care treatments, such as chemotherapy, immunotherapy, or radiation. Such combination approaches have the potential to synergize therapeutic effects, address resistance mechanisms, and improve overall clinical outcomes without imposing excessive systemic toxicity.

Furthermore, utilizing personalized medicine by using Wnt pathway components as biomarkers and thus selecting patients can ensure that treatments are administered to populations most likely to benefit. Finally, advances in drug design, such as small molecules, antibody–drug conjugates, or RNA-based therapies, may offer more effective approaches to targeting Wnt pathway components. These approaches, combined with thorough preclinical validation and well-structured trial designs, could significantly enhance the success rate of Wnt-targeted therapies in GI cancers, ultimately benefiting patients and paving the way for FDA approval after decades of research and trials.

## Figures and Tables

**Figure 1 cells-14-00178-f001:**
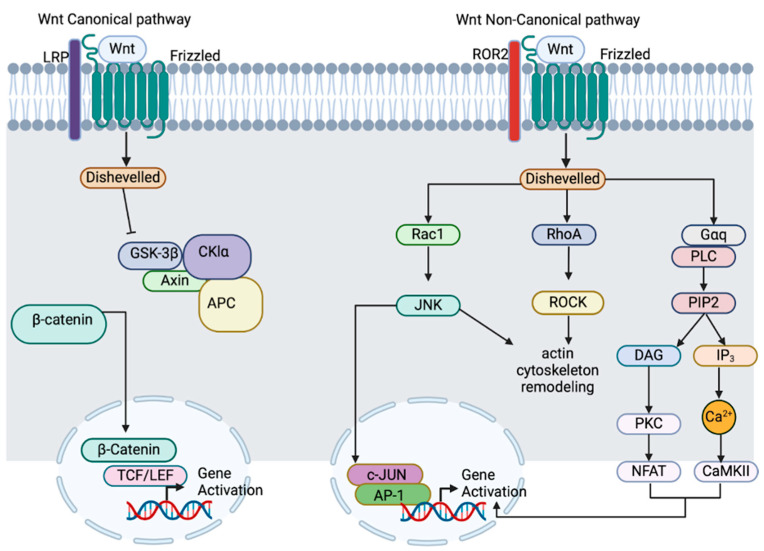
**Canonical Wnt pathway** (**left**): Wnt ligands (such as Wnt1, Wnt2, Wnt3, Wnt3a, and Wnt 8) bind to the frizzled (FZD) receptor and its coreceptors LRP5/6 (LDL-receptor-related proteins). This interaction recruits disheveled (DVL) and disrupts the destruction complex (AXIN, GSK-3β, CKIα, and APC), which normally degrades β-catenin. As a result, β-catenin accumulates in the cytoplasm, translocates to the nucleus, and binds to TCF/LEF transcription factors, leading to gene activation. The target genes of this pathway include important regulators of cellular differentiation and proliferation, such as c-Myc and cyclin D. **Non-canonical pathway** (**right**): The non-canonical Wnt pathways (such as the Wnt/PCP and Wnt/Ca^2^^+^ pathways) are β-catenin-independent and involve different ligands (e.g., Wnt5a, Wnt4, and Wnt11). These ligands bind to FZD and other co-receptors such as ROR2. The signaling cascade activates proteins such as Rac1, RhoA, JNK, ROCK, and Ca^2^^+^-dependent effectors like CAMKII and NFAT. These pathways control processes like cytoskeletal remodeling, cellular migration, and polarization. Figure created on https://BioRender.com accessed on 16 September 2024.

**Figure 2 cells-14-00178-f002:**
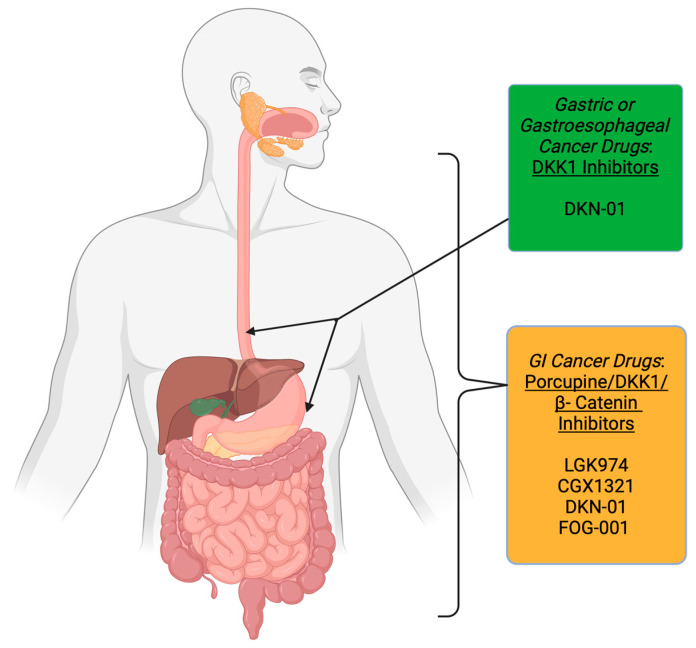
Wnt pathway-targeted drugs in various GI cancers, including gastric and gastroesophageal cancers. The figure illustrates the various Wnt signaling targets and associated drugs that have been or are currently being evaluated in clinical trials for different GI cancers and specifically in gastric and gastroesophageal cancers. Figure created on https://BioRender.com accessed on 3 December 2024.

**Figure 3 cells-14-00178-f003:**
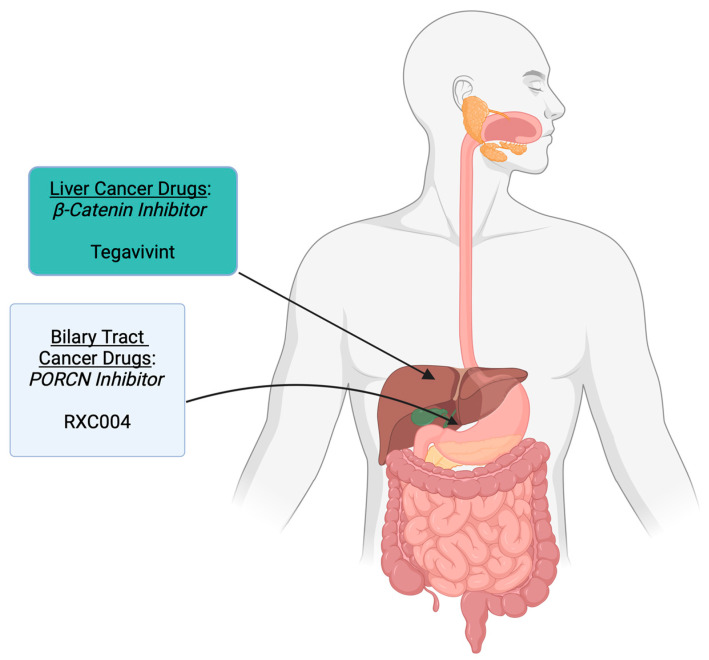
Wnt pathway-targeted drugs in liver cancers. The figure illustrates the various Wnt signaling targets and associated drugs that have been or are currently being evaluated in clinical trials for liver cancers. Figure created on https://BioRender.com accessed on 4 January 2025.

**Figure 4 cells-14-00178-f004:**
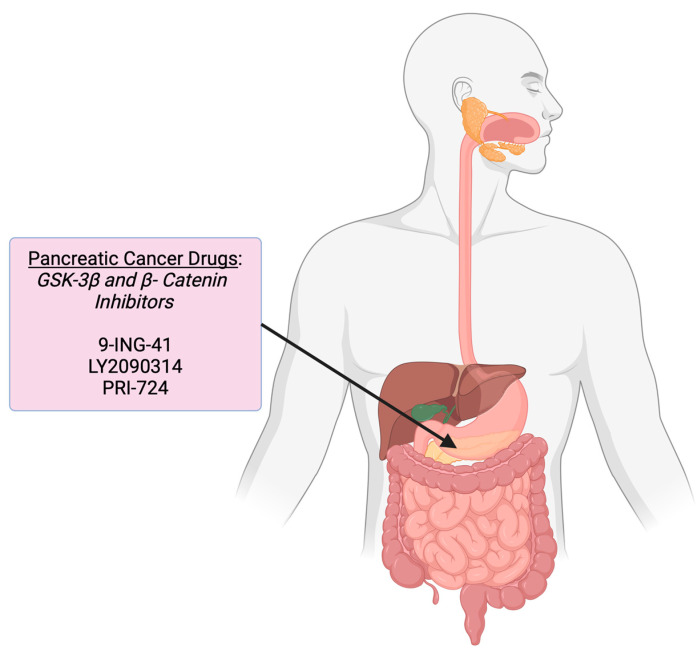
Wnt pathway-targeted drugs in pancreatic cancers. The figure illustrates the various Wnt signaling targets and associated drugs that have been or are currently being evaluated in clinical trials for pancreatic cancers. Figure created on https://BioRender.com accessed on 2 January 2025.

**Table 1 cells-14-00178-t001:** Clinical trials of drugs targeting Wnt signaling pathway in advanced GI cancers that include esophageal and gastric cancers.

Indication	Drug	Target	Regimen	Clinical Status	Phase	Reference
Advanced GI cancers along with other Wnt dependent advanced malignancies	LGK974 (WNT974)	Porcupine	Monotherapy OR Combination with PDR001 (spartalizumab)	Completed	Phase 1	NCT01351103, [94]
Advanced GI tumors	CGX1321	Porcupine	Monotherapy	Unknown	Phase 1	NCT03507998
GI cancers	CGX1321	Porcupine	Monotherapy OR Combination with pembrolizumab (anti-PD-1) OR Combination with encorafenib (targets tyrosine kinases) and cetuximab (anti-EGFR)	Unknown	Phase 1	NCT02675946
Advanced GI cancer along with other advanced solid tumors	DKN-01	DKK1	Combination with the same combination agents used in parent study OR Monotherapy (DKN-01 naïve patients with Wnt activating mutations)	Available *	Intermediate-size Expanded Access Protocol (EAP)	NCT04681248 Parent Study: NCT05480306 (this is phase 2 and is for CRC)
Gastric or gastroesophageal cancer	DKN-01	DKK1	Combination with tislelizumab (anti-PD-1) ± chemotherapy	Active	Phase 2	NCT04363801
Gastric or gastroesophageal cancer	DKN-01	DKK1	Combination with atezolizumab (anti-PD-L1)	Active, not recruiting	Phase 1/2	NCT04166721
Locally advanced or metastatic GI cancers along with other locally advanced or metastatic solid tumors	FOG-001	β-Catenin	Monotherapy	Recruiting	Phase 1/2	NCT05919264

Source: *Clinicaltrials.gov*. Data collected on 24 October 2024. Withdrawn and terminated trials are not included. * Available: Expanded access is currently available for this investigational treatment, and patients who are not participants in the clinical study may be able to gain access to the drug, biologic, or medical device being studied. (Definition from Clinicaltrials.gov Glossary accessed on 24 October 2024).

**Table 2 cells-14-00178-t002:** Clinical trials of drugs targeting Wnt signaling pathway in liver and pancreatic cancers.

Indication	Drug	Target	Regimen	Clinical Status	Phase	Reference
Wnt mutated tumors including hepatocellular carcinoma	Tegavivint (BC2059)	β-catenin/TBL1inhibitor	Monotherapy	Recruiting	Phase 1/2	NCT04851119
Advanced hepatocellular carcinoma	Tegavivint (BC2059)	β-catenin/TBL1inhibitor	Monotherapy AND Combination with pembrolizumab (anti-PD1 antibody)	Recruiting	Phase 1/2	NCT05797805, [123]
Biliary tract cancers	RXC004	Porcupine	Monotherapy OR Combination with nivolumab (anti-PD-1)	Active, Not Recruiting	Phase 1	NCT03447470
Pancreatic adenocarcinoma	Elraglusib (9-ING-41)	GSK-3β	Combination with FOLFIRINOX OR Combination with FOLFIRINOX + losartan (TGF-β blocker)	Recruiting	Phase 2	NCT05077800
Solid tumors including pancreatic adenocarcinoma	Elraglusib (9-ING-41)	GSK-3β	Monotherapy OR Combination with cytotoxic agents including gemcitabine and nab-paclitaxel	Active, Not recruiting	Phase 2	NCT03678883, [124]
Advanced solid tumors	LY2090314	GSK-3β	Monotherapy OR Combination with pemetrexed and carboplatin	Completed	Phase 1	NCT01287520, [125]
Advanced or metastatic pancreatic adenocarcinoma	PRI-724	β-catenin/CBP inhibitor	Combination with gemcitabine	Completed	Phase 1	NCT01764477

Source: Clinicaltrials.gov. Data collected on 24 October 2024. Withdrawn and terminated trials are not included.

## Data Availability

Not applicable.

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
