# Peer review of "Wnt Pathway-Targeted Therapy in Gastrointestinal Cancers: Integrating Benchside Insights with Bedside Applications"

_cells, 2025, doi:10.3390/cells14030178_

Round 1
Reviewer 1 Report
Comments and Suggestions for Authors
Comments
The manuscript with entitled “Wnt Pathway Targeted Therapy in GI Cancers: Integrating 2
Benchside Insights with Bedside Applications”. This topic is highly relevant, addressing the growing interest in Therapy in GI Cancers.
1. Recommendation: Explicitly outline how this review contributes to the field compared to existing literature.
2. There are many cancers related to the Wnt/β-catenin pathway. Recommendation: Expanding the discussion scope of the article.
3. Although the figures and tables in the paper can assist in illustrating some of the content, there is still room for improvement in terms of clarity and completeness. Recommendation: Each section of the article should be illustrated with a diagram to facilitate readers' understanding.
4. Although the article introduces some therapeutic drugs for the Wnt/β-catenin pathway, it is not sufficient. The article should introduce more targeted drugs for clinical treatment and the clinical potential treatment of the Wnt/β-catenin pathway.
5. Recommendation: adding a section specifically discussing current shortcomings and future research priorities and directions.
Author Response
Response to Reviewer 1 Comments:
Thank you very much for taking the time to review this manuscript. Please find the detailed responses below and the corresponding revisions/corrections in red colored font in the re-submitted file.
- Comment 1: Recommendation: Explicitly outline how this review contributes to the field compared to existing literature.
Response 1: We sincerely thank the reviewer for this valuable recommendation. In response, we have added a dedicated paragraph explicitly outlining how this review contributes to the field compared to existing literature. This section can be found on lines 73–83 on page 2. We appreciate the reviewer’s thoughtful feedback and hope this addition meets their expectations.
- Comment 2: There are many cancers related to the Wnt/β-catenin pathway. Recommendation: Expanding the discussion scope of the article.
Response 2: We sincerely thank the reviewer for their valuable recommendation. In response, we have expanded the discussion to provide a broader perspective on the role of the Wnt/β-catenin pathway across various GI cancers. This revised Discussion section, now included alongside the Conclusion (lines 693–704 on page 19), aims to address the reviewer's suggestion. We appreciate the reviewer’s thoughtful feedback and trust this addition strengthens the manuscript.
- Comment 3: Although the figures and tables in the paper can assist in illustrating some of the content, there is still room for improvement in terms of clarity and completeness.
Recommendation: Each section of the article should be illustrated with a diagram to facilitate readers' understanding.
Response 3: We sincerely thank the reviewer for their thoughtful suggestion to enhance the clarity and completeness of our figures. In response, the previous Figure 2, which summarized Wnt pathway-targeted drugs in clinical trials across all GI cancers (Gastric, Esophageal, Liver, and Pancreatic cancers), has been divided into three distinct figures for improved clarity:
- Figure 2: Wnt Pathway-Targeted Drugs in Gastric and Esophageal Cancers (Lines 281–285, Page 8)
- Figure 3: Wnt Pathway-Targeted Drugs in Liver Cancer (Lines 387–390, Page 11)
- Figure 4: Wnt Pathway-Targeted Drugs in Pancreatic Cancer (Lines 387–390, Page 11)
As a result, the previous Figure 3 has been renumbered as Figure 5 (Line 573, Page 16).
Additionally, all in-text references to Figures 3, 4, and 5 have been updated accordingly in the following lines:
- Line 383 (Page 10)
- Lines 415, and 453 (Page 12)
- Lines 459, and 478 (Page 13)
- Line 491 (Page 14)
- Lines 589, 594, 614, 619, and 622 (Page 17)
- Lines 631, 650, 664, and 673 (Page 17)
- Lines 684 and 690 (Page 18)
We believe these changes enhance the visual clarity and accessibility of the data presented and appreciate the reviewer’s valuable input in guiding these improvements.
- Comment 4: Although the article introduces some therapeutic drugs for the Wnt/β-catenin pathway, it is not sufficient. The article should introduce more targeted drugs for clinical treatment and the clinical potential treatment of the Wnt/β-catenin pathway.
Response 4: We sincerely thank the reviewer for their thoughtful comment. To ensure comprehensive coverage, we conducted an exhaustive search of clinical trials targeting the Wnt signaling pathway in gastrointestinal (GI) cancers using ClinicalTrials.gov. Our search strategy included keywords encompassing all key components of the Wnt pathway, its modulators, and specific GI cancer types. Also, preclinical and benchside research studies were identified through PubMed using the same set of Wnt pathway-related terms combined with each GI cancer type. All relevant clinical and preclinical trials obtained through these searches have been included in our review. If there are specific drugs or studies that the reviewer believes we may have overlooked, we would be more than happy to incorporate that information into the manuscript. We greatly appreciate the reviewer’s feedback and commitment to enhancing the quality of our work.
- Comment 5: Recommendation: adding a section specifically discussing current shortcomings and future research priorities and directions.
Response 5: We sincerely thank the reviewer for their thoughtful recommendation. A section specifically addressing current shortcomings and future research priorities and directions has been integrated into the Discussion and Conclusion from lines 706–742 on pages 19 and 20. We appreciate the reviewer’s valuable input and believe this addition strengthens the overall clarity and depth of the manuscript.
Reviewer 2 Report
Comments and Suggestions for Authors
The increased activity of the Wnt pathway is essential for developing various cancers, including gastrointestinal (GI) cancers. From a clinical point of view, it can also be therapeutically important, particularly in creating molecular targeted therapy.
Nayak et al. introduced the canonical and non-canonical Wnt signaling pathways in this review article. Afterward, Wnt pathways were discovered as therapeutic targets in GI cancers: esophageal cancer, gastric cancer, hepatocellular cancer, pancreatic cancer, small intestinal cancer, and colorectal cancer. The available data from clinical trials were presented in the form of three tables.
In my opinion, the paper is worth attention, especially for readers interested in GI tumors and/or targeted therapy. The quality of this article is good, but I suggest several minor corrections:
1. I encourage the authors to prepare a graphical abstract covering the scope of this review article.
2. A full affiliation is needed, not only the name of the University.
3. In keywords, I suggest replacing GI Cancers with Gastrointestinal Cancers and Wnt with the Wnt signaling pathway.
4. In section 2.2. please add the reference to Figure 1 in the main text.
5. Tables 2 and 3 - please edit (expand) the column "Phase" so as not to divide the word "Phase" into two lines.
Author Response
Response to Reviewer 2 Comments:
Thank you very much for taking the time to review this manuscript. Please find the detailed responses below and the corresponding revisions/corrections in red colored font in the re-submitted file.
- Comment 1: I encourage the authors to prepare a graphical abstract covering the scope of this review article.
Response 1: We sincerely thank the reviewer for their valuable suggestion. A graphical abstract summarizing the scope of this review article has been prepared and submitted as per the journal's requirements. We appreciate the reviewer’s recommendation and hope the graphical representation effectively captures the essence of our work.
- Comment 2: A full affiliation is needed, not only the name of the University.
Response 2: We sincerely thank the reviewer for their observation. The full affiliation has been provided in line 6 on page 1 to ensure clarity and alignment with the journal's requirements. We appreciate the reviewer’s attention to detail and hope this revision meets their expectations.
- Comment 3: In keywords, I suggest replacing GI Cancerswith Gastrointestinal Cancers and Wnt with the Wnt signaling pathway.
Response 3: We sincerely thank the reviewer for their thoughtful suggestion. The recommended changes have been implemented in the keywords section on line 23, page 1, replacing "GI Cancers" with "Gastrointestinal Cancers" and "Wnt" with "the Wnt signaling pathway." We appreciate the reviewer’s attention to detail and believe these edits enhance the clarity and precision of the manuscript.
- Comment 4: In section 2.2. please add the reference to Figure 1 in the main text.
Response 4: We sincerely thank the reviewer for their careful observation. A reference to Figure 1 has now been added in line 173 on page 4 in section 2.2. We appreciate the reviewer’s attention to detail and trust this adjustment addresses their concern.
- Comment 5: Tables 2 and 3 - please edit (expand) the column "Phase"so as not to divide the word "Phase" into two lines.
Response 5: We sincerely thank the reviewer for their attention to detail. Tables 2 and 3 have been reformatted and expanded, ensuring that the word "Phase" now appears on a single line for improved clarity and presentation. These changes have been applied on pages 6, 10, and 15. We appreciate the reviewer’s suggestion and hope the revised tables meet their expectations.
Reviewer 3 Report
Comments and Suggestions for Authors
Dear Authors
This manuscript was well organized and have good information about Wnt Pathway Targeted Therapy in GI Cancers in detail.
Here are minor comments
1. The authos should contain "Discussion" besides "Conclusion".
2. Please avoid displaying the table in a narrow format. Instead, display it in full-screen mode to minimize word breaks.
3. Please, check the style of Cells including email address, and Author Contributions, etc.
4. Please, draw the chemical structure of drugs in all Tables.
Author Response
Response to Reviewer 3 Comments:
Thank you very much for taking the time to review this manuscript. Please find the detailed responses below and the corresponding revisions/corrections in red colored font in the re-submitted file.
- Comment 1: The authors should contain "Discussion" besides "Conclusion.”
Response 1: We sincerely thank the reviewer for their insightful suggestion. A Discussion section has now been included alongside the Conclusion, spanning lines 692–742 on pages 19 and 20. We appreciate the reviewer’s valuable feedback and trust this revision enhances the clarity and completeness of the manuscript.
- Comment 2: Please avoid displaying the table in a narrow format. Instead, display it in full-screen mode to minimize word breaks.
Response 2: We sincerely thank the reviewer for their valuable suggestion. All tables have been reformatted and expanded to ensure they are displayed in a full-screen layout, minimizing any word breaks for improved readability and clarity. We appreciate the reviewer’s attention to this detail and hope the revised presentation meets their expectations.
- Comment 3: Please check the style of Cells, including email address, Author Contributions, etc.
Response 3: We sincerely thank the reviewer for their valuable feedback. The email address in the correspondence section (Line 8, Page 1) has been removed and adjusted to align with the Cells journal style. Additionally, the Author Contributions section (Lines 744–749, Page 20) has been added according to the journal's guidelines. We appreciate the reviewer’s attention to detail and trust that these adjustments address their concerns.
- Comment 4: Please, draw the chemical structure of drugs in all Tables.
Response 4: We sincerely thank the reviewer for their thoughtful suggestion. While we understand the value of including chemical structures for clarity, many of the drugs discussed, such as antibiotics and natural derivatives, possess highly complex and large structures that would be challenging to represent effectively within the tables. To maintain consistency, we have opted not to include chemical structures for any of the listed compounds. We hope this rationale is acceptable and appreciate the reviewer’s understanding.
Round 2
Reviewer 1 Report
Comments and Suggestions for Authors
None